# Human papillomavirus vaccination practices and perceptions among Ghanaian Healthcare Providers: A qualitative study based on multi-theory model

Peter Agyei-Baffour[ID][1], Matthew Asare[2]*, Beth Lanning[2], Adofo Koranteng[1], Cassandra Millan[2], Mary E. Commeh[3], Jane R. Montealegre[4], Hadii M. Mamudu[5]

1 School of Public Health, Kwame Nkrumah University of Science and Technology, Kumasi, Ghana, 2 Department of Public Health, Baylor University, Waco, Texas, United States of America, 3 Non-Communicable Disease Control, Ghana Health Services, Accra, Ghana, 4 Dan L Duncan Comprehensive Cancer Center, Baylor College of Medicine Houston, Houston, Texas, United States of America, 5 Department of Health Services Management and Policy, College of Public Health, East Tennessee State University, Johnson City, Tennessee, United States of America

* matt_asare@baylor.edu

**Data Availability Statement:** All relevant data are within the manuscript and its Supporting Information file.

## Abstract

### Background

Healthcare providers' (HCPs) recommendations for the Human Papillomavirus (HPV) vaccine are likely to increase the vaccination uptake. However, little is known about Ghanaian HCPs' general practices regarding HPV vaccination. We used Multi-Theory Model (MTM) constructs (i.e. participatory dialogue, behavioral confidence, environment, social and emotional transformation) to examine Ghanaian HCPs' attitudes towards HPV vaccination and their vaccination recommendation practices.

### Methods

We conducted three, 60-minute focus group discussions (FGDs) with HCP in the second-largest government hospital in Ghana. Sixteen semi-structured open-ended questions based on MTM constructs were used to guide the FGDs. We explored HCPs' general knowledge about HPV, vaccination recommendation behavior, physical environment, and socio-cultural factors associated with the HPV vaccination. Data from the FGDs were transcribed and thematically coded using NVivo software.

### Results

The sample of (n = 29) HCPs consisting of males (n = 15) and females (n = 14) between the ages of 29 and 42 years participated in the FGDs. Our analyses showed that HCPs (a) rarely offered HPV vaccination recommendations, (b) showed varied understanding about who should be vaccinated regarding age eligibility, gender, and infection status. Perceived barriers to HPV vaccination include (a) low urgency for vaccination education due to competing priorities such as malaria and HIV/AIDS; (b) lack of data on HPV vaccination; (c) lack of awareness about the vaccine safety and efficacy; (c) lack of HPV vaccine accessibility

**Funding:** The author(s) received no specific funding for this work.

**Competing interests:** The authors have declared that no competing interests exist

and (d) stigma, misconceptions and religious objections. HCPs expressed that their motivation for counseling their clients about HPV vaccination would be increased by having more knowledge about the vaccine's efficacy and safety, and the involvement of the parents, chiefs, churches, and opinion leaders in the vaccination programs.

## Conclusion

The study's findings underscore the need for a comprehensive HPV vaccination education for HCPs in Ghana. Future HPV vaccination education programs should include information about the efficacy of the vaccine and effective vaccination messages to help mitigate HPV vaccine-related stigma.

## Introduction

Human Papillomavirus (HPV) causes several cancers, including cervical, anal, penile, and oropharyngeal cancer, as well as genital warts and recurrent respiratory papillomatosis [1]. HPV-related cancers exert disproportionate human suffering and economic burden on Low- and Middle-Income countries (LMICs) including Ghana. The HPV related cancers represent unmet medical needs in Ghana, where there were over 3,300 new cases of HPV-related cancers and 2,977 HPV-related cancer deaths in 2018 [2]. HPV-related cancers accounted for almost 20% of cancer deaths in Ghana [2]. These data show that HPV diseases are a menace to public health in LMICs in general and Ghana in particular.

HPV vaccines (Cervarix®, Gardasil®, and Gardasil 9®) have strong and demonstrated efficacy for protecting against HPV-related diseases in other countries. The availability of these vaccines has been associated with a decrease in HPV related cancers including cervical dysplasia, the precursor of cervical cancer [3–6]. The World Health Organization (WHO) recommends two doses of vaccination for youths between the ages of 9 and 14 years old [3,7–10]. However, three doses are recommended for both males and females aged 15 through 26 years who have not been previously vaccinated [11]. In Ghana, the HPV vaccination was first introduced in 2013 [12] and currently, both Cervarix® and Gardasil®, are sold in the Ghana market. However, in Ghana, like many LMICs, the vaccination rates remain very low [13]. In 2013, the Global Alliance for Vaccines and Immunizations (GAVI) [14,15] and the GARDA-SIL® Access Program (GAP) [15], successfully pilot-tested HPV vaccination programs in Ghana. Since then there is a limited effort to increase the HPV vaccination uptake in Ghana. A compounding challenge is the lack of HPV vaccination uptake data in Ghana, [16–18] making it difficult to target priority areas and evaluate vaccination programs. Currently, in Ghana HPV vaccination is not a government mandated program. Ghana health services (GHS) are responsible for the distribution of the HPV vaccination in the communities. Besides, a few designated polyclinics (e.g. Komfo Anokye Teaching Hospital in Kumasi) and hospitals (e.g. Suntresu Hospital in Kumasi) are responsible for women's health which includes HPV vaccination. There is no health insurance coverage for HPV vaccination (M., Commeh, personal communication, July 20, 2019).

Studies in other countries showed that Healthcare providers (HCPs) including physicians, nurses, and allied health workers play a central role in HPV vaccine uptake [19–21]. HCPs' recommendations, referral, reminders or/and counseling for the vaccine is/are associated with parental decisions to vaccinate their child [22] and the general increase in the vaccination uptake [23,24]. Research in other settings indicates that multiple factors influence HCPs' HPV

vaccination recommendation practices including HCPs' perceived effectiveness of the HPV vaccines in reducing cervical cancer [25], HCPs' perceived behavioral control, self-efficacy, outcome expectancy and confidence to address parents' concerns [22,26], and the value placed on professional organizations recommendations [25,27,28]. However, little is known about Ghanaians HCPs' beliefs and attitudes towards the vaccination. Here we identified factors that influence Ghanaian HCPs' general practices including general knowledge about HPV, vaccination recommendation behavior, physical environment, and socio-cultural factors about HPV vaccinations. We also examined HCPs' perceptions of factors that could encourage parents and adolescents to participate in HPV vaccination. Understanding Ghanaian HCPs' predisposing, enabling and reinforcing factors and their general professional practices regarding the HPV vaccine is important for developing evidence-based interventions to improve the consistency and strength of HPV vaccine recommendation behaviors among HCPs. We applied the Multi-theory Model of behavior change (MTM) to gain an understanding of Ghanaian HCPs' HPV vaccination practices.

### Theoretical framework

Focus group development and analysis were guided by the multi-theory-model (MTM) of behavior change. This theoretical framework has two main components: initiation of the behavior changes and continuation of the health behavior change. The initiation of behavior change includes; participatory dialogue (individuals weigh advantages versus disadvantages), behavioral confidence, and changes in the physical environment [29]. The continuation or sustenance of behavior involves; emotional transformation, practice for change, and changes in the social environment [29]. Theoretically based studies to understand Ghanaian HCPs' attitudes and behavior towards HPV vaccination are limited and this study seeks to close the gap in the literature.

## Methods

### Study setting and participants

We recruited the HCPs in May 2020 from Ghana Health Services (GHS) and the Komfo Anokye Teaching Hospital, the second-largest government hospital in Kumasi, in the Ashanti Region of Ghana. A purposive sampling strategy was used in the recruitment processes to ensure a diversity of perspectives based on HCPs' specialty and departmental settings. The inclusion criteria were a healthcare provider (physician, nurse, hospital administrator) who was 18 years or above, worked at GHS and any HPV vaccination designated poly clinics and hospitals in Kumasi in the Ashanti Region, able to speak English, and was willing to participate in the study. The inclusion of health professionals other than those who have direct contacts with patients is necessary for obtaining HPV vaccination information from diverse perspectives. Exclusion criteria were providers (physician, nurse, hospital administrator) who do not work in any of the HPV vaccination designated hospitals or clinic in Kumasi, Ashanti Region. We recruited HCP participants through face-to-face contacts and verbal invitations at the hospital and GHS in Kumasi. The research field coordinator went to the GHS (they are officially responsible to go to the communities to vaccinate adolescents) office in Kumasi in the Ashanti Region, met with individuals in their offices and introduced the study to them. After questions from the prospective participants about the study and clarification by the research field coordinator, the prospective participants who met the inclusion criteria and indicated willingness to participate were given a specific date and time for the focus group discussions. We recruited participants from the poly clinics (Komfo Anokye Teaching Hospital through the field coordinator visiting the midwifery departments (the designated entities in the clinic to administer

HPV vaccination), introducingthe study to the doctors and nurses, and gaving a verbal invitation to them to join the focus group. The participants were also given the specific date and time for the focus group discussions. On the agreed dates and times, the field coordinator physically went back to remind the invited health professionals from both GHS and the hospital of the focus group discussions.

## Sampling

Glaser and Strauss [30] suggested that sample size for qualitative study should be determined by saturation point. However, Bertaux [31] argued that a minimum of 15 participants is an acceptable sample size for a qualitative study. In the current study conscious efforts were made to recruit 29 health professionals from GHA and Komfo Anokye Teaching Hospital so as to gather more representative data.

## Ethical considerations

The study's protocol was approved by the Institutional Review Boards (IRBs) of Baylor University and Kwame Nkrumah University of Science and Technology in Ghana. We asked the participants to provide oral consent in this study. No written consent was used in this study because we considered the use of oral consent as a culturally appropriate way of obtaining consent [32,33]. Had either of the IRBs objected to the use of oral consent, we would have considered the written consent but both IRBs agreed oral consent was appropriate At the beginning of each of the focus group discussions, the principal investigator read the informed consent aloud to the participants, gave them a chance to ask any questions or voice concerns about participating in the study, and they asked the participants to indicate their willingness to participate by responding "yes" or "no." In the end, each of the invited participants provided oral consent (by responding yes) before joining the study. Informed consent form, which was read orally, is available in the supporting information file.Data collection procedure.

   We conducted three, sixty-minute, focus group discussions with HCPs. The focus group discussions were conducted by the field coordinator (AD) and the principal investigator (MA). We did not conduct a sample size or power calculation. Based on a literature review [30,31], we used saturation point as the determinant for sample size. Ahead of the focus groups, participants answered a brief demographic survey indicating age, gender, education, marital status, insurance status, religion and job title. All the focus group discussions were conducted at the Komfo Anokye Teaching Hospital research library. For the focus groups, each participant was assigned a number, sat in a circle and answered questions in a round-robin format, that is, the participants taking turns to respond to the questions. The discussion guide consisted of sixteen semi-structured open-ended questions based on MTM constructs, with additional follow-up questions added for clarifications. We elicited information about participatory dialogue (individuals weigh advantages versus disadvantages), behavioral confidence, changes in the physical environment, emotional transformation, practice for change, and changes in the socio-cultural factors associated with the HPV vaccination from the HCPs. Additionally, we explored general knowledge about HPV vaccination and HCPs' vaccination recommendation behaviors. All the discussions were audio-recorded with a hand-held digital recorder and they were independently transcribed verbatim by two of the researchers (MA and AA).

## Data analysis

Data were evaluated through directed content analysis [34], which is utilized when a theory is used to understand a phenomenon. The process includes identifying key concepts or variables

as initial coding categories and operational definitions for each category are determined using the theoretical constructs. Two of the researchers (MA and CM) independently read the transcripts and highlighted all text that on the first impression appeared to represent the MTM constructs. We coded all the highlighted passages using the predetermined codes based on the MTM constructs including participatory dialogue behavioral confidence, changes in the physical environment, emotional transformation, practice for change, and changes in the socio-cultural factors. We gave a new code to any text that could not be categorized with the initial coding scheme. We came together and discussed our individual codes and created a consensus list of preliminary codes. Preliminary coded transcripts were discussed in an iterative manner to ensure general agreement and subcategories (themes) were created and coded under each of the MTM constructs using NVivo 12 Plus. We resolved any disagreement through discussions and each theme was reported with supporting direct quote(s). In this manuscript, quotes are summarized in a table. In the text, we denote a *number (e.g. 1)* and a letter (e.g. a) where the number represents a subheading for the MTM construct and the letter represents themes identified within each subheading. The number and the letter referred to in the text denote the direct quotes in the table.

## Results

### Demographic characteristics

Of the total participants in the study (n = 29), 31.03% were physicians, 27.59% nurses, 17.24% immunization field officers, and 24.14%other healthcare professionals (biostatisticians, pharmacists, hospital administrator, etc.). Half of the participants were (51%) were males. The mean age of the participants 35.24 ± 4.10 years. Participants with undergraduate degrees comprised a slight majority (51.72%). The basic demographic characteristics of the participants are described in Table 1.

### Main findings

The main discussions centered on knowledge about the HPV virus and the vaccination. The interview themes within the Multi-Theory Model constructs and participants' direct quotes are presented in **Table 2**.

**Knowledge about HPV.** The HCPs were aware of HPV transmission mode and its related health consequences (Table 2, quote *1a*). It was commonly understood among the groups that HPV is sexually transmitted, and that HPV virus causes genital warts and cancers (example anal, cervical, oral). However, the HCPs did not know about the health burden of HPV cases in Ghana because they reported that there is no epidemiological data indicating the prevalence, incidence, morbidity, and mortality of HPV related diseases. They indicated that HPV-related diseases are aggregated with other viral-related diseases and that is part of the reasons why data are not available on HPV health burden in Ghana (*1b*).

**Knowledge about HPV vaccinations.** We assessed the HCPs' general knowledge about HPV vaccinations including the recommended series of doses, the target population (male and/or female) and age eligibility and the vaccine importance. The HCPs' response showed varied understanding about who should be vaccinated especially about at what age eligibility, gender, and infection status (positive vs. negative) were the main factors in the response variations. The HCPs' responses about the target population for HPV vaccination included only girls aged 11–12 years old, only immunocompromised persons, and everyone even including children. They indentified HPV sreening as a pre-condition for HPV vaccination. Three doses were nearly unanimously recognized as the typical quantity of doses for the HPV vaccine (*2a-2e*).

**Table 1. Demographic characteristics of the participants (n = 29).**

| Variable | Frequency | Percentage |
|---|---|---|
| **Age** | **Mean = 35.24** | **SD = 4.10** |
| **Gender** | | |
| Male | 15 | 51.72 |
| Female | 14 | 48.28 |
| Marital Status | | |
| Married | 21 | 72.41 |
| Never married | 8 | 27.59 |
| **Education** | | |
| High School | 1 | 3.5 |
| Undergraduate | 15 | 51.72 |
| Graduate | 13 | 44.83 |
| **Annual Income** | | |
| <20,000 | 10 | 34.48 |
| 20,000–30,000 | 16 | 55.17 |
| 35,000–50,000 | 3 | 10.34 |
| **Religion** | | |
| Christianity | 27 | 93.10 |
| Other | 2 | 6.90 |
| **Insurance** | | |
| Yes | 100 | 100 |
| No | 0 | 0 |
| **Profession** | | |
| Physician | 9 | 31.03 |
| Nurses | 8 | 27.59 |
| Immunization field officer | 5 | 17.24 |
| Other | 7 | 24.14 |

**Initiation of communication.** Two themes emerged under HCPs' general practices of initiating HPV vaccination communication and recommendation. The first theme is education and counseling during a family planning center visit (*3a-3b*). HCPs in the family care center at the hospital reported that they include the HPV vaccination education for an adolescent only when they know that adolescent is sexually active. Again, they counseled patients about HPV vaccination only when patients are seeking treatment for other reproductive related care or when a patient is showing symptoms of HPV infections.

The second theme was limited or no HPV vaccination communication due to the competing health priorities (*3c-3f*). HCPs explained that they concentrate on the acute health issues and have limited time for preventive care. Other barriers include HPV vaccination is less priority because of lack of awareness and evidence to support efficacy and safety, and/or unavailability of resources (immunization kits), and lack of evidence to support the association between the HPV infection and cancer. Furthermore, HCPs reported that they do not recommend HPV vaccination because, for the most part, health care in the low-and middle-income countries like Ghana is episodic and related to acute illness.

**Participatory dialogue.** HCPs identified several reasons (advantages) that could encourage the HPV vaccination uptake (*4a-4b*). They include personal beliefs that the HPV vaccine prevents HPV infection, resulting in financial saving (prevention cost vs. treatment cost), building the adolescent antibodies to help them build their immune system against HPV infections, reducing morbidity and mortality associated with HPV related diseases.

**Table 2. Interview themes within the multi-theory model constructs, and participants' quotes.**

| Constructs | Themes | Quote |
|---|---|---|
| *1. HPV Knowledge* | (a) Transmission | "I have heard about the HPV virus as a type of virus that can cause cervical cancer and also other types of cancer. I also heard that it can be transmitted sexually. And also, during childbirth, it can also be transmitted from the mother to the infant." (Group 1, Participant 2). |
| | (b) Lack of Data | "We don't have data specifically on HPV. I think once it is viral, it mimics characteristics of any other viral so I think we just lump them together and reporting is a problem so if education goes up and the reporting patterns changes then we can report on it really." (Group 2, participant 5) "Few studies have been done in this area and the other side is that people are not actually looking for it to quantify the disease burden." (Group 2, participant 4) |
| *2. HPV Vaccination Knowledge* | (a)Target population | "Every woman who is sexually active or every young girl who is growing up should be vaccinated to prevent it. It is not only who is sexually active in some way you might get it in another way or another form which are probably they may have a small percentage so somehow they might be vaccinated; everybody must be vaccinated." (Group 2, participant 1). "People with immunocompromising conditions like people living with HIV (Group 1, participant 7). "Ladies or girls between the ages of 11 and 12 are most advised to get the vaccinations, so at their adolescent age.(Group 1, participant 6)" "…The focus should be on the youth, when you analyze sexual activeness with age categorization, you will see that youth dominates so if the vaccination can target the youth I think it will be proper." (Group 2, participant 5) |
| | (b) Schedule dose | "The vaccination is called Gardasil and Cervarix. And it is given 3 doses, or the recommendation is that 3 doses should be given. The first dose is given within the first month and the second month you will be given the next dose and six months from the first dose then you will be given the third dose. That is how they do it. (Group 1, participant 9) |
| | (c) Infectious status | "Anybody who is tested and is negative can be vaccinated" (Group 2, participant 9) "I don't understand what you are saying because you can't just vaccinate someone without doing the pap smear first,. how are you going to make sure that the person is positive or negative. They need to be tested first before the vaccine and if they are positive we don't give them the vaccine" (Group 1, participant 8). |
| | (d) Age eligibility (e) Vaccine importance | "I support the initial concept that people could be vaccinated at birth, in that way once you are born and you have immunity to the HPV. At birth, you have the vaccination to protect." (Group 2, participant 4) "So teenagers have become sexually active, so if we are to vaccinate, it's better to start from the secondary schools, even possibly the junior secondary schools here." (Group 3, participant 1) "It's important because it protects for a lifetime. If you are being vaccinated with three doses, it means you are protected from the HPV virus for a lifetime." (Group 1, participant 2)"When you are being infected by this virus, the likelihood of you getting cancer is high. So it is very important to get vaccinated to stay away from cancer and all the others." (Group 3, participant 4) |
| *3. Initiation of communication* | | |
| Education & Counseling | | |
| | (a) Hospital visit | "… those of us at the family planning center, when someone comes there, we normally do education, health education for them. … We talk about it as the virus, how it occurs and how it causes cervical cancer and then the pap smear as well." (Group 1, participant 9). |
| | (b) Positive status | "The pap smear will let the person go and get the vaccination, because if the person is sexually active then they have to be tested before the vaccine. If they tested positive, they don't get vaccination" (Group 1, participant 9). |
| Limited Vaccine Comm. | | |
| | (c) Neglect | "It's one of the conditions that we said it is neglected, we don't focus on it. Because we don't have data and I don't know what it is public health importance to authority. So once we don't have an interest in that aspect even education and awareness may not be done." (Group 2, participant 6) "Sometimes the problem that we are facing in the family planning now, people come in to do the pap smear but we don't have the kits for seven months to almost a year, we don't have the kits. Look at this. We don't have the kits for them to test whether they are negative or positive before they go for the vaccine." (Group 1, participant 9) |
| | (d) Competing priorities | "It is not our priority. We have some diseases like malaria, HIV, TB that we focus on that. Like Ebola, those that can cause pandemic or epidemic, the rest but for HPV it's not our priority so we even don't talk about it." (Group 2 participant 7) |
| | (e) No data | "… if HPV is the lead contributory factor or is the only way one can get Cervical cancer, then our data is not correct. …, we are using the ICD (International Classification of Disease) data to code all the ailments that are reported here. So, if the doctors catering for such cases lump everything up and write CL of the service, without giving us all the other contributing diagnosis or ailments …, then we also not report on HPV. So, data in African and Ghana for that matter is something." (Group 2, participant 5). "HPV I would say we don't know much or we don't have much data on it. If we do it's silent. It's not like malaria or tuberculosis whereby we can just click and get it." (Group 1, participant 1) |

*(Continued)*

**Table 2.** (Continued)

| Constructs | Themes | Quote |
|---|---|---|
| | (f) Acute illness | *"I think it's because of the clinical situation. You know when the patients come your aim is to address the condition they present with; so if it doesn't go in that direction, it's not likely that you bring that topic up." (Group 3, participant 2).* |
| **4. Participatory dialogue** | | |
| Advantages | | |
| | (a) Build immune system | *"Your antibodies will be developed, and your immune system will be stronger to fight the virus." (Group 2, participant 5)* |
| | (b) Prevention | *"From a social scientist's point of view, the vaccination, as you take it, it prevents the disease rather than to wait for you to acquire the disease and then start treatment which is expensive." (Group 1, participant 1) "To protect you from a disease that is preventable. It is also to, you know, it protects the community depending on the number of people who are vaccinated, even if someone comes into the system with the disease." (Group 3, participant 2)* |
| Disadvantages | | |
| | (c) Stigma | *You see, the stigma that comes with it, especially the things that- there is this notion that once you go for this vaccine it means that you have the intention of either engaging in sexual activities or already doing so, people stay away because of the stigma associated with it." (Group 3, participant 10) "Some females wouldn't want their peers to know they are sexually active or they want to become sexually active. So in order not to let them know, they wouldn't even go for it in the first place." (Group 3, participant 3)* |
| | (d) Religious objection | *"Talking about churches, most churches will not encourage pre-marital sex. So they may see it as a means of promoting immorality when you vaccinate adolescents and teenagers, they are seeing you as amoral and they will not support it because of that." (Group 3, participant 4).* |
| | (e) Discomfort/pain with injection | *"Every vaccine has its own way of making people uncomfortable or has its own side effects. It may make you feel weak." (Group 2, participant 1) "Not all patients like to be injected, in fact, the majority of patients would avoid injections as much as possible . . . after the vaccination, it comes with so much pain that the arm will hurt for quite a number of days so discomfort, it will deter people from getting vaccinated." (Group 3, participant 1)* |
| **5. Behavioral Confidence** | (a) Predisposing | *"We need to have access to information, how safe the drug is, the studies that have been done to evaluate the safety and efficacy aspect of the drugs to convince myself that it is safe and efficacious, that is when I will give it to others. So, the information about safety and side effects should be transparent." (Group 2, participant 4)* |
| | (b) Enabling | *"The parent consent will make me feel confident. The involvement of the parent in the child's health will give me the confidence to give HPV vaccination." (Group 2, participant 2).* |
| | | *"I thinks it should be encouraged from the national level where there will be education from the national level about the seriousness of the HPV and its health implications. Like HIV people are aware of its health implications because of the involvement of national leaders and awareness created by the national media. So, if we show on the TV the problem associated with HPV and the problem with cancer, I think people will become aware. In most cases, people died of cervical cancer and they are not aware, and the family member will be saying that the person was just sick and went to the hospital and just died but if we create awareness, people will see the seriousness of the problems." (Group 2, participant 3)* |
| | (c) Reinforcing | *"And, other people have come for the vaccine and it has worked very well for them. So that will give much confidence in delivering any information about the vaccines to them." (Group 2, participant 3) "Everything is based on evidence and I can cite an issue whereby there is no evidence that the vaccine gives complications or anything. So it is based on evidence. Even though many of them have been administered, you haven't heard of any complications coming out of that so that confidence level is high and the vaccine is effective." (Group 1, participant 1)* |
| **6. Physical Environment** | (a) Structural | *"The availability of the vaccines. If I educate somebody and if the vaccine is not there what are we going to do? So, the vaccines should be available at reasonable prices before we even talk about education. (Group 2, participant 2) "When you are able to get to these points of vaccination, you need to at least travel from one point to the other. Your means of transport one and also the available roads and then the other means to get to where you are going to do the vaccination is important because some people may be hindered because they have to travel a long dusty road and by the end of the day they will be- their clothes will be filled with dust and they will be tired; so they will not be encouraged to even attend even if you were to organize a vaccination." (Group 3, participant 7)* |
| | (b) Monetary | *"I want to look at if from that the government will buy the vaccines and make it available to the ordinary Ghanaians. The government can buy it, reduce the price or subsidize the price to let's say 5 Ghana Cedis,[less than $1.00 equivalent] that will help others to go for the vaccinations.Simply put, there is no widespread, systematic, publicly available HPV vaccination program in Ghana to increase vaccination rates" (Group 2, participant 5).* |

(*Continued*)

**Table 2.** (Continued)

| Constructs | Themes | Quote |
|---|---|---|
| | (c) Administrative | *"We need Facilities, personnel, monetary incentives, education and refresher courses. We need the human personnel before you can initiate any move. Without human personnel, you cannot build the facilities and the whatever. If you don't have human personnel, it would be very difficult to champion any course." (Group 1, participant 9)* |
| *7. Social Environment* | (a) Informational and tangible support | *"We need support in the community and parents, we need ethical clearance (parental approval). . . . I would say the social support that we need is parental contribution or support. And if we are doing it in the community, we need to get one or 2 opinion leaders and explain for them to understand the issue. So if we receive support from the community leaders it will help boost our morale to continue to offer education but before we receive those opinion leaders' support they need to understand what the vaccination is all about. If community leaders provide a platform by mobilizing the people for us to provide education to the masses." (Group 1, participant 1)* |
| | | *"In our communities, we hold the churches in high esteem so to involve them to be able to advocate for the vaccine would be an advantage for the program." (Group 3, participant 7).* |
| *8. Sustenance* | | *"After I have given the injection, I will take the parents' phone number so that . . ., I will give them a call, . . . to remind them that their child is due for their next injection. . . give them a small book to write the next date for injection in it." (Group 1, participant 9).* |

The disadvantages include discomfort, pain with injection, unwanted side effects, religious objection because it encourages engaging in immoral behaviors, and promoting promiscuity objection, infertility, and stigma associated with the vaccination because others will assume that individuals who are sexually active are the ones who will go for the HPV vaccine (4c-4e).

**Behavioral confidence.** On the HCPs' sources of confidence for counseling or educating teenagers and parents about the vaccination, three themes emerged. They were predisposing, enabling and reinforcing factors. The predisposing factors include HCPs' knowledge and awareness about the efficacy of the vaccination (*5a*). HCPs mentioned that knowledge of the source of the vaccine, efficacy of the vaccine, safety profile of the vaccine, and public awareness of the seriousness of HPV could improve their confidence level for providing it to the client. The enabling factors that could increase HCPs' confidence in HPV vaccine education include parental, community and national leaders' involvement and support (*5b*). The participants indicated parents' involvement in the decision to vaccinate their children would motivate them to give vaccinations or offer education. The level of moral support and trust HCPs receive from the community and national leaders could influence HCPs to do more about HPV vaccination and education. Reinforcing theme derived from the discussions include previous success from the HPV vaccination. Participants indicated that they would have the confidence to administer or talk to parents and adolescents about the vaccination if they know that the vaccination works (*5c*).

**Physical environment.** Environmental factors that encourage or discourage the promotion of vaccine or vaccine education include structural and administrative supports. The structural factors include monetary (insurance and the cost of the vaccination), vaccines availability, facilities, good roads and means of transportation to deliver the vaccine to the community, incentives (snacks and refreshments) to attract people to vaccination centers, and no widespread, systematic, publicly available HPV vaccination program in Ghana to increase vaccination rates (6a-6b). The administrative variables include personnel, financial support, and refresher education courses or additional training, availability of current information about the vaccination, a reliable registry for HPV vaccination, and no national vaccination program, (6c).

**Social environment.** HCPs indicated that the kind of social support needed includes informational and tangible supports. Informational support includes parental approval. (7a). They need support and involvement of opinion leaders including, assemblymen, pastors,

imams, and chiefs in leading the HPV vaccination education efforts. This would boost the HCPs' morale to provide vaccine education and mobilize the community. However, they perceived that church involvement may be difficult since HPV vaccination may be thought of as encouraging immoral behaviors. The HCPs reported that the use of media such as radio, TV programs, and church gatherings could encourage them and the community for HPV vaccines. The tangible support they need include personnel, training and workshops, HPV vaccination kits, monetary incentives, and physically mobilizing the people in their communities for vaccinations.

**Sustenance.** The methods HCPs indicated the methods used for ensuring completion of the series of vaccination include the use of appointment reminder cards, phone calls a week ahead of the next visit (*8*). They recommended the use of the motivated staff to volunteer (just like patient navigators) to go to the communities to remind the people about their due dates.

## Discussion

HPV-related cancers represent unmet medical needs in Ghana. HCPs' role to facilitate HPV vaccination to address this medical need cannot be underestimated. However, the attitudes and behaviors of Ghanaian HCPs towards HPV vaccines have been understudied. Therefore, guided by the multi-theory model (MTM), this study examined factors that influence the attitudes and behaviors of Ghanaian HCPs towards HPV vaccination. Additionally, we examined HCPs' knowledge about HPV vaccination, general recommendation/referral practices, and how MTM constructs explain their general HPV vaccination practices. The key findings included: general knowledge about HPV, gaps in knowledge about HPV vaccination and its influence on HCPs' HPV vaccination recommendation and education behaviors, and general factors for increasing HPV vaccination in Ghana.

Similar to Hoque et al study among South African doctors, our findings show the HCPs have a general understanding of the HPV mode of transmission and its health implications [35]. Our study showed the HCPs have some basic knowledge about the HPV vaccination but there were varied views and concerns about the recommended age eligibility for vaccination, the availability, efficacy, and the safety of the vaccine. This finding supports Farias (2018) study among Pediatricians who expressed concerns about HPV vaccine safety and efficacy [36]. Our finding of lack of general HPV vaccination recommendations and counseling practice by HCPs is consistent with Suryadevara et al [37] study that concluded that pediatricians in their study did not routinely recommend HPV vaccine to eligible patients. In particular, the concept of "medical home" in which medical care is coordinated by a primary care physician is not well integrated into the healthcare system in Ghana and many LMICs. Vaccines are provided through vaccination clinics or mobile clinics that are run by the government and delivered to communities. Therefore, HPV vaccination educations or referrals are normally done by a few designated clinics. HCPs in our study also identified several other factors that impede delivery of the HPV vaccine, including the competing health priorities such as malaria and HIV; lack of surveillance data documenting the prevalence of the HPV related cancers, lack of awareness about the causal relationship between persistent HPV infection and cervical cancer, and confusion about the age eligibility and dosing schedule account for their low HPV vaccination counseling or recommendation in Ghana. Several studies including a systematic review [38] have come to a similar conclusion that HCPs' low knowledge of the vaccine [27,39–41], and low-self-confidence [27,42] about HPV vaccination correspond with low vaccine recommendation rate [38]. Similarly, our study finding agrees with a few other studies that have argued that confusing about age eligibility [28,43,44] and dosing schedule [45] contribute to HCPs' unwillingness to recommend or counsel parents and adolescents about the vaccination

[38]. Another interesting finding is that HCPs reported the use of the HPV vaccination screening as a precondition for HPV vaccination. Individual's positive status indicates that the person will not be vaccinated but only those who tested negative would be vaccinated. However, in the high income countries like the US, the vaccine is recommended for individuals who are within the specific age eligibility, regardless of the HPV infectious status [6,46] and that testing or screening for HPV status is not recommended [47].

Multi-Theory Model (MTM) constructs were used to determine the factors that could encourage HPV vaccination uptake. The *Participatory dialogue* construct discusses the reasons for (advantages) and against (disadvantages) HPV vaccination. HCPs stated that the advantages of HPV vaccination include HPV virus prevention, the building of the immune system, and the reduction of morbidity and mortality rates. They identified that the disadvantages of HPV vaccination include discomfort, pain with injection, unwanted side effects, encourage engaging in immoral behaviors and promoting promiscuity. Our findings support the conclusions of many studies that indicated that discomfort [41,48] and concerns that teens will practice risky sexual behavior affect their recommendation behaviors [37,38,42].

*The behavioral confidence* construct discusses factors that would build HCPs' confidence to recommend HPV vaccination. The key behavioral confidence factors HCPs identified include an increase in awareness about the efficacy and safety of the vaccination, parental, community and national leaders' involvement and support for the vaccination. These findings underscore that parents, community or opinion and national leaders support increases the health care HCPs' confidence in the general practices of HPV vaccination recommendations [22,49,50].

*The social environment* construct determines social factors that could enhance the HCPs' willingness to recommend HPV vaccination. Similar to factors identified under the behavioral confidence construct, HCPs reported the involvement of opinion leaders including assemblymen, pastors, imams, and chiefs could increase the vaccination uptake and this finding substantiates Schmidt-Grimminger et al [41] study's that opinion leaders e.g. fathers and grandparents could be influential in the HPV vaccination decision making. HCPs also recognized that the use of communication channels such as Media radio, TV programs, and church gatherings could encourage them. This finding agrees with Dilley et al [48] findings that the use of media to educate could encourage HPV vaccinations.

*The physical environment* construct examines physical factors in the environment that could contribute to the HCPs' HPV vaccination recommendation practices. Consistent with several studies, HCPs in our study identified monetary (insurance and the cost of the vaccination) factors, vaccines availability, good roads and means of transportation to deliver the vaccine to the community, personnel [26,36,48] and refresher educational courses or additional training [22,51], as physical environmental factors that could increase the HPV vaccination.

*Sustenance construct* discusses the best practices that HCPs could use to get parents and adolescents to complete the recommended series of the HPV vaccination. The HCPs in our study listed phone calls, text messages and sending volunteers to remind parents and adolescents could increase HPV vaccination. Hudson et al studies show that HCPs use of reminders such as text messages, phone calls, and prompts can improve HPV vaccine completion rates, and some HPV vaccination interventions that included parental reminders have succeeded [52].

## Limitations and strengths

The goal of this study was to gain a more in-depth understanding of HCPs' perceptions of HPV vaccination and provide insight into ways the provider may be able to increase vaccine

uptake. We acknowledge that the types of HCPs who participated in our focus groups limits the generalizability of our findings. Specifically, HCPs were recruited from the Komfo Anokye Teaching Hospital and the Ghana Health Services, and their perceptions may not be representative of other HCPs, such as those working in rural settings and community clinics Additionally, inclusion of the health professional such as health administrators, biostatisticians and pharmacists, who do not directly provide service to patients was a limitation because for the most part they were not able to contribute well when we were discussing communication of HPV vaccination to parents and adolescents. The face-to-face nature of the focus group discussion presents several challenges including socially desirable responses. Another limitation of focus groups is the potential for the domination of the discussions by the most outspoken HCPs, and uncomfortable feeling from some of the introvert HCPs and may refuse to talk about their experiences. To control for possible socially desirable responses, we assured the participants that their individual responses would be kept in strict confidence and all study information would be de-identified. We conducted the focus group discussions in a round-robin format to give every participant the opportunity to talk.

Despite the noted limitations, our findings provide valuable insight into the HPV vaccination process in Ghana. First, this study helps us to determine the challenges of and motivations for HPV vaccination from low- and middle-income country HCPs who are not only underrepresented in the discourse of the HPV vaccination but also are hard to reach individuals. Second, the use of MTM to guide the study is important because MTM constructs—participatory dialogue [53], behavioral confidence [54,55], social environment, physical environment [54,56], and emotional transformation [57] have been empirically tested and proven to be efficacious in guiding studies. Finally, the inclusion of other health care professionals (administrators, biostatisticians, pharmacist) other than providers with direct contact with patients shows the strength of the study for two reasons. (a) This study is first of its kind in Ghana and the administrators, biostatisticians, pharmacist provided information about the logistical, administrative, and facilities challenges to HPV vaccination in Ghana. For instance, the biostatisticians discussed the lack of epidemiological data (i.e. incidence, prevalence) on HPV infection and cervical cancer cases, the pharmacist contributed immensely about the concern of the efficacy, safety and the availability of the vaccine (b) The inclusion of community fieldworkers from Ghana Health Services provided perspectives from an institution which is in charge of the distribution of HPV vaccination in Ghana. The community fieldworker provided much information about going to communities to provide vaccinations to adolescents and the kind of challenges they face.

## Implications

The findings of this study have several implications. We observed divergent views about HPV vaccination age eligibility, schedule doses, the target populations, and the HPV infection status. This finding underscores the need for HCPs-specific education and communication training. Specifically, interventions program or HPV vaccination educational training on the World Health Organization recommendations for HPV vaccinations that include two doses of vaccination for 9 to 14-year-olds [3,7–10] and three doses for both males and females aged 15 through 26 who have not been previously vaccinated [11]. HCPs' knowledge about the safety of the vaccine is low so intervention to increase knowledge about the safety and efficacy of the vaccine and to equip HCPs with strategies to communicate about the vaccine could help build the confidence of HCPs to counsel parents and adolescents. Proper surveillance data documenting the prevalence and incidence of the disease could be used to advance an argument to support the reprioritization of HPV vaccination as an important public health priority. To

address the stigma about the vaccination, an intervention to reframe the conversation about the vaccination as cancer prevention may help. This was underscored by our findings that HCPs often discussed the sexual nature of HPV, tied the initiation of the vaccine series to sexual debut, and expressed that parents of their patients were concerned with the potential sexual implications of the vaccine. An intervention designed to increase HPV vaccination should include the involvement of churches, the political and community leaders to counter the current narrative connecting the HPV vaccine and sexual behavior. MTM constructs including participatory dialogue, behavioral confidence, changes in the physical environment, social environment, and sustenance could be used to guide intervention to educated HCPs in Ghana.

## Conclusion

The findings of our study underscore the need for a comprehensive HPV vaccination educational or intervention program in Ghana. The program should be designed to (a) equip health care HCPs and the community with knowledge about the efficacy of HPV vaccination to prevent cancer, (b) address transportation issues for both vaccination delivery and access, and (c) address HPV vaccination-related stigma to increase vaccination uptake. This comprehensive program is critical as the health workers will be armed with the right information to educate the public, dispel misconceptions and promote the uptake of the vaccine. Ultimately, this will be a giant step forward in Ghana's attempt at cervical cancer prevention and control.

## Supporting information

**S1 File. Combined.**
(DOCX)

## Author Contributions

**Conceptualization:** Peter Agyei-Baffour, Matthew Asare, Beth Lanning, Jane R. Montealegre, Hadii M. Mamudu.

**Data curation:** Peter Agyei-Baffour, Matthew Asare, Adofo Koranteng, Mary E. Commeh.

**Formal analysis:** Peter Agyei-Baffour, Matthew Asare.

**Investigation:** Peter Agyei-Baffour.

**Methodology:** Peter Agyei-Baffour, Matthew Asare, Adofo Koranteng.

**Project administration:** Adofo Koranteng.

**Supervision:** Peter Agyei-Baffour, Matthew Asare, Beth Lanning, Mary E. Commeh.

**Validation:** Peter Agyei-Baffour, Matthew Asare, Cassandra Millan, Jane R. Montealegre.

**Writing – original draft:** Matthew Asare, Mary E. Commeh.

**Writing – review & editing:** Peter Agyei-Baffour, Matthew Asare, Beth Lanning, Adofo Koranteng, Cassandra Millan, Mary E. Commeh, Jane R. Montealegre, Hadii M. Mamudu.

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
