## [Decision Letter · Decision Letter 0]

28 Aug 2020

PONE-D-20-06397

Human Papillomavirus Vaccination Practices and Perceptions among Ghanaian Healthcare Providers: A Qualitative Study Based on Multi-Theory Model .

PLOS ONE

Dear Dr. Agyei-Baffour,

Thank you for submitting your manuscript to PLOS ONE. After careful consideration, we feel that it has merit but does not fully meet PLOS ONE’s publication criteria as it currently stands. Therefore, we invite you to submit a revised version of the manuscript that addresses the points raised during the review process.

Both of the reviewers have returned extremely positive reviews about your paper, however, they have both suggested minor revisions to improve the paper, including further details which provide context to the delivery of HPV vaccination in the country, and around the justification for the approach taken in the study. Please attempt to address all of the issues raised. 

We look forward to receiving your revised manuscript.

Kind regards,

Holly Seale

Academic Editor

PLOS ONE

Journal Requirements:

2.Thank you for stating that the participants provided verbal consent. In the Methods, please clarify that participants provided oral consent. Please also state in the Methods:

- Why written consent could not be obtained

- Whether the Institutional Review Board (IRB) approved use of oral consent

- Whether consent was informed

- How oral consent was documented

For more information, please see our guidelines for human subjects research: https://journals.plos.org/plosone/s/submission-guidelines#loc-human-subjects-research.

3.Please provide a sample size and power calculation in the Methods, or discuss the reasons for not performing one before study initiation.  

4. In your Methods section, please provide additional information about the participant recruitment method and the demographic details of your participants. Please ensure you have provided sufficient details to replicate the analyses such as: a) the recruitment date range (month and year), b) a description of any inclusion/exclusion criteria that were applied to participant recruitment, c) a description of how participants were recruited, and d) descriptions of where participants were recruited and where the research took place.

5.We note that you have indicated that data from this study are available upon request. PLOS only allows data to be available upon request if there are legal or ethical restrictions on sharing data publicly. For more information on unacceptable data access restrictions, please see http://journals.plos.org/plosone/s/data-availability#loc-unacceptable-data-access-restrictions.

Reviewers' comments:

Reviewer's Responses to Questions

**Comments to the Author**

1. Is the manuscript technically sound, and do the data support the conclusions?

Reviewer #1: Yes

Reviewer #2: Partly

2. Has the statistical analysis been performed appropriately and rigorously? 

Reviewer #1: N/A

Reviewer #2: N/A

3. Have the authors made all data underlying the findings in their manuscript fully available?

Reviewer #1: Yes

Reviewer #2: No

4. Is the manuscript presented in an intelligible fashion and written in standard English?

Reviewer #1: Yes

Reviewer #2: Yes

5. Review Comments to the Author

Reviewer #1: Hello, this is a great paper, I enjoyed reading it. I had a few minor comments. One, I would like to see a brief discussion in the Introduction providing some background on how the HPV vaccine is currently distributed across Ghana. Does the Ghana MoH have a cervical cancer/HPV vaccine country plan (seems not)? Is there a partnership with GAVI to shoulder part of the vaccine cost? Is there a national health insurance program? I am wondering if there is a national policy or program for vaccine distribution, or the absence of, that might explain low uptake of the vaccine, in addition to the other factors listed. Two, in the Methods section, can you provide more information on your inclusion criteria for the sample of participants. HCPs were targeted, but was the study limited to those who provide direct care to patients? Was there a minimum training/education threshold? Demographics are provided in the Tables, but some additional description of the inclusion/exclusion criteria in the Methods would help. Finally, if possible, can you ensure that each subtheme has 2 illustrative quotes? It would help build the breadth of the data presented to the reader.

Reviewer #2: The purpose of this study was to understand HPV vaccine practices and perceptions among healthcare providers in Ghana. This strong manuscript is timely, relevant and will be of interest to readers. One improvement needed is to clarify results and discussion based on the fact that not all participants were physicians with direct patient contact or ability to recommend/give vaccine.

Introduction: Cite opening sentence.

Last sentence in Theoretical Framework is confusing (studies are understudied)

What type of healthcare providers were recruited? Not stated until the results, but were there inclusion/exclusion criteria? What is the value in recruiting participants that do not directly give vaccinations (administrators, biostatisticians, etc.). There are varying expectations of what a physician should know about HPV versus an administrator. Justify this- need to understand perceptions at multiple levels of the healthcare team, etc. A focus only on provider/patient communication is not sufficient when this is the sample.

Most of themes are surrounding provider patient communication but not all participants were providers or would have any patient contact. How do you justify this/how robust are the themes? How do we know what information came from those interacting with patients vs. those who do not? This may affect the integrity of the theme if the thoughts came from participants who will never be communicating with a patient. Also applies to the behavioral confidence aspect of the theory used.

Table 2 is useful and well organized

6. PLOS authors have the option to publish the peer review history of their article (what does this mean?). If published, this will include your full peer review and any attached files.

Reviewer #1: No

Reviewer #2: No

---

## [Author Response · Author response to Decision Letter 0]

25 Sep 2020

Dear Editor,

REF: Human Papillomavirus Vaccination Practices and Perceptions among Ghanaian Healthcare Providers: A Qualitative Study Based on Multi-Theory Model. 

We are thankful for the opportunity to revise our manuscript in response to the reviewers’ comments. We have carefully read the reviewers comments and modified the manuscript to reflect the suggestions and comments the reviewers provided. All changes from the original submission are tracked in the manuscript with the file name 'Revised Manuscript with Track Changes.' Besides, we have provided a clean copy of the revised version of the manuscript and it is saved under a file name 'Manuscript'

Our responses to the reviewers' comments are provided below in red font.

Thank you very much for your time. Please let me know if you have any questions.

Yours

Peter Agyei-Baffour

 

Comments from the reviewers

 Formatted it per PLOS ONE’s requirements

2.Thank you for stating that the participants provided verbal consent. In the Methods, please clarify that participants provided oral consent. Please also state in the Methods:

- Why written consent could not be obtained

- Whether the Institutional Review Board (IRB) approved use of oral consent

- Whether consent was informed

- How oral consent was documented

For more information, please see our guidelines for human subjects research: https://journals.plos.org/plosone/s/submission-guidelines#loc-human-subjects-research.

 We addressed this consent with the following statements. We also provided a copy of the informed consent form. Also see page 7 under method section: Ethical Considerations

Now it reads

“The study’s protocol was approved by the Institutional Review Boards (IRBs) of Baylor University and Kwame Nkrumah University of Science and Technology in Ghana. We asked the participants to provide oral consent in this study. No written consent was used in this study because we considered the use of oral consent as a culturally appropriate way of obtaining consent [32, 33]. Had either of the IRBs objected to the use of oral consent, we would have considered the written consent but both IRBs agreed oral consent was appropriate At the beginning of each of the focus group discussions, the principal investigator read the informed consent aloud to the participants, gave them a chance to ask any questions or voice concerns about participating in the study, and they asked the participants to indicate their willingness to participate by responding “yes” or “no.” In the end, each of the invited participants provided oral consent (by responding yes) before joining the study. Informed consent form, which was read orally, is available in the supporting information file.”

3.Please provide a sample size and power calculation in the Methods or discuss the reasons for not performing one before study initiation. 

We addressed this issue with the following statements. Also see page 6 under method section: Sampling

Now it reads 

“Sampling

Glaser and Strauss[30] suggested that sample size for qualitative study should be determined by saturation point. However, Bertaux [31] argued that a minimum of 15 participants is an acceptable sample size for a qualitative study. In the current study conscious efforts were made to recruit 29 health professionals from GHA and Komfo Anokye Teaching Hospital so as to gather more representative data.”

4. In your Methods section, please provide additional information about the participant recruitment method and the demographic details of your participants. Please ensure you have provided sufficient details to replicate the analyses such as:

a) the recruitment date range (month and year), 

Addressed this comment. See page 5 under method section

Now it reads 

“We recruited the HCPs in May 2020 from Ghana Health Services (GHS) and the Komfo Anokye Teaching Hospital,…”

b) a description of any inclusion/exclusion criteria that were applied to participant recruitment,

Addressed this comment. See page 5 under method section

Now it reads

“The inclusion criteria were a healthcare provider (physician, nurse, hospital administrator) who was 18 years or above, worked at GHS and any HPV vaccination designated poly clinics and hospitals in Kumasi in the Ashanti Region, able to speak English, and was willing to participate in the study. Exclusion criteria were providers (physician, nurse, hospital administrator) who do not work in any of the HPV vaccination designated hospitals or clinic in Kumasi, Ashanti Region”. 

c) a description of how participants were recruited, and d) descriptions of where participants were recruited and where the research took place.

Addressed this comment. See pages 5 through 7 under method section

Now it reads

“Study setting and participants

We recruited the HCPs in May 2020 from Ghana Health Services (GHS) and the Komfo Anokye Teaching Hospital, the second-largest government hospital in Kumasi, in the Ashanti Region of Ghana. A purposive sampling strategy was used in the recruitment processes to ensure a diversity of perspectives based on HCPs’ specialty and departmental settings. The inclusion criteria were a healthcare provider (physician, nurse, hospital administrator) who was 18 years or above, worked at GHS and any HPV vaccination designated poly clinics and hospitals in Kumasi in the Ashanti Region, able to speak English, and was willing to participate in the study. Exclusion criteria were providers (physician, nurse, hospital administrator) who do not work in any of the HPV vaccination designated hospitals or clinic in Kumasi, Ashanti Region. We recruited HCP participants through face-to-face contacts and verbal invitations at the hospital and GHS in Kumasi. . The research field coordinator went to the GHS (they are officially responsible to go to the communities to vaccinate adolescents) office in Kumasi in the Ashanti Region, met with individuals in their offices and introduced the study to them. After questions from the prospective participants about the study and clarification by the research field coordinator, the prospective participants who met the inclusion criteria and indicated willingness to participate were given a specific date and time for the focus group discussions. We recruited participants from the poly clinics (Komfo Anokye Teaching Hospital through the field coordinator visiting the midwifery departments ( the designated entities in the clinic to administer HPV vaccination), introducingthe study to the doctors and nurses, and gaving a verbal invitation to them to join the focus group. The participants were also given the specific date and time for the focus group discussions. On the agreed dates and times, the field coordinator physically went back to remind the invited health professionals from both GHS and the hospital of the focus group discussions.

Sampling

Glaser and Strauss[30] suggested that sample size for qualitative study should be determined by saturation point. However, Bertaux [31] argued that a minimum of 15 participants is an acceptable sample size for a qualitative study. In the current study conscious efforts were made to recruit 29 health professionals from GHA and Komfo Anokye Teaching Hospital so as to gather more representative data.

Ethical Considerations

.The study’s protocol was approved by the Institutional Review Boards (IRBs) of Baylor University and Kwame Nkrumah University of Science and Technology in Ghana. We asked the participants to provide oral consent in this study. No written consent was used in this study because we considered the use of oral consent as a culturally appropriate way of obtaining consent [32, 33]. Had either of the IRBs objected to the use of oral consent, we would have considered the written consent but both IRBs agreed oral consent was appropriate At the beginning of each of the focus group discussions, the principal investigator read the informed consent aloud to the participants, gave them a chance to ask any questions or voice concerns about participating in the study, and they asked the participants to indicate their willingness to participate by responding “yes” or “no.” In the end, each of the invited participants provided oral consent (by responding yes) before joining the study. Informed consent form, which was read orally, is available in the supporting information file.

Data collection procedure

We conducted three, sixty-minute, focus group discussions with HCPs. The focus group discussions were conducted by the field coordinator (AD) and the principal investigator (MA). We did not conduct a sample size or power calculation. Based on a literature review [30,31], we used saturation point as the determinant for sample size. Ahead of the focus groups, participants answered a brief demographic survey indicating age, gender, education, marital status, insurance status, religion and job title. All the focus group discussions were conducted at the Komfo Anokye Teaching Hospital research library. For the focus groups, each participant was assigned a number, sat in a circle and answered questions in a round-robin format, that is, the participants taking turns to respond to the questions. The discussion guide consisted of sixteen semi-structured open-ended questions based on MTM constructs, with additional follow-up questions added for clarifications. We elicited information about participatory dialogue (individuals weigh advantages versus disadvantages), behavioral confidence, changes in the physical environment, emotional transformation, practice for change, and changes in the socio-cultural factors associated with the HPV vaccination from the HCPs. Additionally, we explored general knowledge about HPV vaccination and HCPs’ vaccination recommendation behaviors. All the discussions were audio-recorded with a hand-held digital recorder and they were independently transcribed verbatim by two of the researchers (MA and AA).”

5.We note that you have indicated that data from this study are available upon request. PLOS only allows data to be available upon request if there are legal or ethical restrictions on sharing data publicly. For more information on unacceptable data access restrictions, please see http://journals.plos.org/plosone/s/data-availability#loc-unacceptable-data-access-restrictions.

We have included data from the study. See the supporting information file 

 Addressed see above comment

Addressed see above comment

Addressed see above comment

 5. Review Comments to the Author

Reviewer #1: Hello, this is a great paper, I enjoyed reading it. 

It is so encouraging to know that you like it. Thank you very much for this feedback. 

I had a few minor comments. One, I would like to see a brief discussion in the Introduction providing some background on how the HPV vaccine is currently distributed across Ghana. Does the Ghana MoH have a cervical cancer/HPV vaccine country plan (seems not)? Is there a partnership with GAVI to shoulder part of the vaccine cost? Is there a national health insurance program? I am wondering if there is a national policy or program for vaccine distribution, or the absence of, that might explain low uptake of the vaccine, in addition to the other factors listed. 

Thank you for your suggestions: We have addressed it by inserting the following statements in the introduction. Also see pages 3 and 4 under introduction section.

“Currently, in Ghana HPV vaccination is not a government mandated program. Ghana health services (GHS) are responsible for the distribution of the HPV vaccination in the communities. Besides, a few designated polyclinics (e.g. Komfo Anokye Teaching Hospital in Kumasi) and hospitals (e.g. Suntresu Hospital in Kumasi) are responsible for women’s health which includes HPV vaccination. There is no health insurance coverage for HPV vaccination (M., Commeh, personal communication, July 20, 2019).”

Two, in the Methods section, can you provide more information on your inclusion criteria for the sample of participants. HCPs were targeted, but was the study limited to those who provide direct care to patients? Was there a minimum training/education threshold? Demographics are provided in the Tables, but some additional description of the inclusion/exclusion criteria in the Methods would help.

Thank you very much for this suggestion. We have added inclusion and exclusion criteria in the method section. See pages 5 and 6 of the manuscript. We have also addressed under point 4 of this rebuttal letter. 

Finally, if possible, can you ensure that each subtheme has 2 illustrative quotes? It would help build the breadth of the data presented to the reader.

Thank you very much for this observation. We have provided two or more quotes in Table 2 to support each subtheme. Providing quotes again in the text will be redundant and it will take more space. If we delete table 2, then providing quotes for each subtheme will be extremely important. We think reviewer#2 agrees that “Table 2 is useful and well organized” so we will maintain the table format.

Reviewer #2: The purpose of this study was to understand HPV vaccine practices and perceptions among healthcare providers in Ghana. This strong manuscript is timely, relevant and will be of interest to readers. 

We are thankful for these comments. Exciting to know that.

One improvement needed is to clarify results and discussion based on the fact that not all participants were physicians with direct patient contact or ability to recommend/give vaccine.

Thank you very much for this comment. Reading through your other comments, we noticed that you have given detailed explanation about your concern on this topic. So our response to this concern can be found below

Introduction: Cite opening sentence.

Thank you very much for this comment. we have cited the opening sentence. See page 1

Last sentence in Theoretical Framework is confusing (studies are understudied)

We have clarified this statement. See page 5 before method section.

 Now it reads

“Theoretically based studies to understand Ghanaian HCPs’ attitudes and behavior towards HPV vaccination are limited and this study seeks to close the gap in the literature.” 

What type of healthcare providers were recruited? Not stated until the results, but were there inclusion/exclusion criteria?

Thank you very much for this suggestion. We have added inclusion and exclusion criteria in the method section. See pages 5 and 6 of the manuscript. We have also addressed under point 4 of this rebuttal letter. 

 What is the value in recruiting participants that do not directly give vaccinations (administrators, biostatisticians, etc.). There are varying expectations of what a physician should know about HPV versus an administrator. Justify this- need to understand perceptions at multiple levels of the healthcare team, etc. A focus only on provider/patient communication is not sufficient when this is the sample.

Most of themes are surrounding provider patient communication but not all participants were providers or would have any patient contact. How do you justify this/how robust are the themes? How do we know what information came from those interacting with patients vs. those who do not? This may affect the integrity of the theme if the thoughts came from participants who will never be communicating with a patient. Also applies to the behavioral confidence aspect of the theory used.

Thank you very much for these thought provoking questions. We have addressed these concerns in the method and discussion sections. See page 5 and 6 of the method section and page 24 of the discussion section.

In the method section, we added the following statement

“The inclusion of health professionals other than those who have direct contacts with patients is necessary for obtaining HPV vaccination information from diverse perspectives.”

In the discussion section, we added the following statement

“Finally, the inclusion of other health care professionals (administrators, biostatisticians, pharmacist) other than providers with direct contact with patients shows the strength of the study for two reasons. (a) This study is first of its kind in Ghana and the administrators, biostatisticians, pharmacist provided information about the logistical, administrative, and facilities challenges to HPV vaccination in Ghana. For instance, the biostatisticians discussed the lack of epidemiological data (i.e. incidence, prevalence) on HPV infection and cervical cancer cases, the pharmacist contributed immensely about the concern of the efficacy, safety and the availability of the vaccine (b) The inclusion of community fieldworkers from Ghana Health Services provided perspectives from an institution which is in charge of the distribution of HPV vaccination in Ghana. The community fieldworker provided much information about going to communities to provide vaccinations to adolescents and the kind of challenges they face.”

Table 2 is useful and well organized

 It is so encouraging to know that you like it. Thank you very much for this feedback

---

## [Editor Report · Decision Letter 1]

1 Oct 2020

Human Papillomavirus Vaccination Practices and Perceptions among Ghanaian Healthcare Providers: A Qualitative Study Based on Multi-Theory Model .

PONE-D-20-06397R1

Dear Dr. Agyei-Baffour,

We’re pleased to inform you that your manuscript has been judged scientifically suitable for publication and will be formally accepted for publication once it meets all outstanding technical requirements.

Kind regards,

Holly Seale

Academic Editor

PLOS ONE
---

## [Editor Report · Acceptance letter]

5 Oct 2020

PONE-D-20-06397R1 

Human Papillomavirus Vaccination Practices and Perceptions among Ghanaian Healthcare Providers: A Qualitative Study Based on Multi-Theory Model. 

Dear Dr. Agyei-Baffour:

I'm pleased to inform you that your manuscript has been deemed suitable for publication in PLOS ONE. Congratulations! Your manuscript is now with our production department. 

Kind regards, 

on behalf of

Dr. Holly Seale 

Academic Editor

PLOS ONE